# Rif2 protects Rap1-depleted telomeres from MRX-mediated degradation in *Saccharomyces cerevisiae*

**Fernando Rodrigo Rosas Bringas[1], Sonia Stinus[1], Pien de Zoeten[1], Marita Cohn[2], Michael Chang[1]***

[1]European Research Institute for the Biology of Ageing, University Medical Center Groningen, Groningen, Netherlands; [2]Department of Biology, Lund University, Lund, Sweden

**Abstract** Rap1 is the main protein that binds double-stranded telomeric DNA in *Saccharomyces cerevisiae*. Examination of the telomere functions of Rap1 is complicated by the fact that it also acts as a transcriptional regulator of hundreds of genes and is encoded by an essential gene. In this study, we disrupt Rap1 telomere association by expressing a mutant telomerase RNA subunit (tlc1-tm) that introduces mutant telomeric repeats. *tlc1-tm* cells grow similar to wild-type cells, although depletion of Rap1 at telomeres causes defects in telomere length regulation and telomere capping. Rif2 is a protein normally recruited to telomeres by Rap1, but we show that Rif2 can still associate with Rap1-depleted *tlc1-tm* telomeres, and that this association is required to inhibit telomere degradation by the MRX complex. Rif2 and the Ku complex work in parallel to prevent *tlc1-tm* telomere degradation; *tlc1-tm* cells lacking Rif2 and the Ku complex are inviable. The partially redundant mechanisms may explain the rapid evolution of telomere components in budding yeast species.

## Editor's evaluation

The study clarifies the role of Rif2 in telomere homeostasis and how cells can extend telomeres and control senescence in the absence of the Rap1 binding to telomeres. The possibility of coping with telomere sequence modification through flexibility and redundancy of capping proteins is of general interest in terms of telomere evolution.

*For correspondence: m.chang@umcg.nl

Competing interest: The authors declare that no competing interests exist.

## Introduction

Telomeres, nucleoprotein complexes located at the ends of eukaryotic chromosomes, protect chromosome ends from degradation, from telomere-telomere fusion events, and from being recognized as double-stranded DNA breaks (*Wellinger and Zakian, 2012*; *de Lange, 2018*). In most eukaryotic species, telomeric DNA consists of tandem G/C-rich repeats of double-stranded DNA with the G-rich strand extending to form a 3′ single-stranded overhang. These repeats are bound by specialized proteins—some to the double-stranded region and others to the 3′ overhang—which are important for proper telomere function. Telomere length is maintained by a dynamic process of shortening and lengthening. Telomeres shorten due to incomplete DNA replication and nucleolytic degradation, and are lengthened by the action of a specialized reverse transcriptase called telomerase (*Wellinger, 2014*). At its core, telomerase consists of a catalytic protein subunit and an RNA subunit, and extends telomeres by iterative reverse transcription of a short G-rich sequence to the 3′ overhang, using the RNA subunit as a template.

Rap1 is the main double-stranded telomeric DNA-binding protein in the budding yeast *Saccharomyces cerevisiae* (*Buchman et al., 1988*; *Conrad et al., 1990*), with important roles in regulating telomere length (*Lustig et al., 1990*; *Conrad et al., 1990*; *Marcand et al., 1997*), transcriptional silencing of subtelomeric genes (*Kyrion et al., 1993*), and preventing telomere-telomere fusions (*Pardo and Marcand, 2005*). Rap1 mediates these functions in part by recruiting additional proteins (i.e., Rif1, Rif2, Sir3, and Sir4) via its C-terminal domain (*Hardy et al., 1992*; *Wotton and Shore, 1997*; *Moretti et al., 1994*). Expression of a mutant Rap1 lacking the C-terminal domain, which retains the ability to bind telomeric sequences, mimics the deletion of the *RIF* and *SIR* genes in terms of telomere length regulation and subtelomeric gene silencing (*Wotton and Shore, 1997*; *Kyrion et al., 1993*). However, the telomeric functions of Rap1 are not limited to the recruitment of Rif and Sir proteins. For example, although Rif2 and Sir4 are important for the inhibition of telomere end-to-end fusions, Rap1 also inhibits fusions independently of Rif2 and Sir4 (*Marcand et al., 2008*). In addition, binding of Rap1 to telomeric repeats is thought to promote replication fork pausing and breakage of dicentric chromosomes at telomere fusions independently of the Rif and Sir proteins (*Makovets et al., 2004*; *Guérin et al., 2019*; *Douglas and Diffley, 2021*). Despite the central role of Rap1 in yeast telomere biology, its study is complicated by the fact that Rap1 is also a transcription factor that regulates the expression of a few hundred genes (*Lieb et al., 2001*). Moreover, *RAP1* is essential for viability (*Shore and Nasmyth, 1987*), preventing analysis of a gene deletion mutant.

In this study, we examined the function of Rap1 at *S. cerevisiae* telomeres by expressing a mutant telomerase RNA subunit (tlc1-tm) that introduces $[(TG)_{0-4}TGG]_n ATTTGG$ mutant telomeric repeats instead of wild-type $(TG)_{0-6}TGGGTGTG(G)_{0-1}$ repeats (*Chang et al., 2007*; *Förstemann and Lingner, 2001*). We find that Rap1 binds very poorly to mutant telomeric sequences, yet *tlc1-tm* cells are viable and grow similar to wild-type cells. The depletion of Rap1, not surprisingly, causes telomere length homeostasis defects. Unexpectedly, the overall levels of Rif2 at telomeres are unaffected. Rif2 recruitment to *tlc1-tm* telomeres is dependent on the MRX complex, and Rif2 is crucial for preventing *tlc1-tm* telomeres from degradation by the MRX complex. The yeast Ku complex functions in parallel to protect *tlc1-tm* telomeres, and absence of both Rif2 and the Ku complex renders *tlc1-tm* cells inviable. Our findings reveal multiple redundant mechanisms that may have been important for the rapid evolution of telomere components in budding yeasts.

## Results

### Rap1 binds poorly to *tlc1-tm* sequences

To examine the function of Rap1 specifically at telomeres, we made use of a telomerase RNA mutant, *tlc1-tm*. Rap1 has a consensus DNA-binding sequence of 5'-CACCCAYACMYM-3' (where Y is C or T, and M is A or C) containing an invariable CCC core (*Graham and Chambers, 1994*; *Lieb et al., 2001*). The template region of wild-type TLC1 is 3'-CACACACCCACACCAC-5', resulting in the addition of $(TG)_{0-6}TGGGTGTG(G)_{0-1}$ telomeric repeats (*Förstemann and Lingner, 2001*). The tlc1-tm template region is 3'-CACCUAAACCACACAC-5', resulting in $[(TG)_{0-4}TGG]_n ATTTGG$ mutant telomeric repeats (*Chang et al., 2007*). The lack of the GGG motif in the mutant repeat sequence should disrupt Rap1 binding, yet the *tlc1-tm* mutant grows similar to a wild-type strain (*Figure 1A*). Telomeres in the *tlc1-tm* mutant are on average longer and more heterogeneous in length than in wild-type strains (*Figure 1B*), but the telomere profile of *tlc1-tm* is much less dramatically altered compared to most other *TLC1* mutants with altered template sequences (*Förstemann et al., 2003*; *Lin et al., 2004*). Telomeres in *tlc1-tm* cells were previously reported to be slightly shorter than in wild-type cells, as measured by telomere PCR (*Chang et al., 2007*). This discrepancy is likely due to the tendency of PCR to amplify shorter fragments more efficiently than longer ones.

Because telomerase only adds repeats to the distal end of telomeres, the proximal region of *tlc1-tm* telomeres still contains some wild-type repeats with the capacity to be bound by Rap1. Thus, we engineered strains in which the left arm of chromosome VII (VII-L) ends with a telomere that consists entirely of either wild-type or *tlc1-tm* telomeric sequence (*Figure 1C*). These strains were constructed by replacing the native VII-L telomere with the *URA3* gene followed by 81 bp of wild-type or 84 bp of *tlc1-tm* telomeric seed sequence, which was lengthened in vivo by either wild-type or *tlc1-tm* mutant telomerase, respectively. As expected, the VII-L telomere containing only mutant sequence (VII-L-MUT) was longer and more heterogeneous than the wild-type control telomere (VII-L-WT) (*Figure 1C*). We

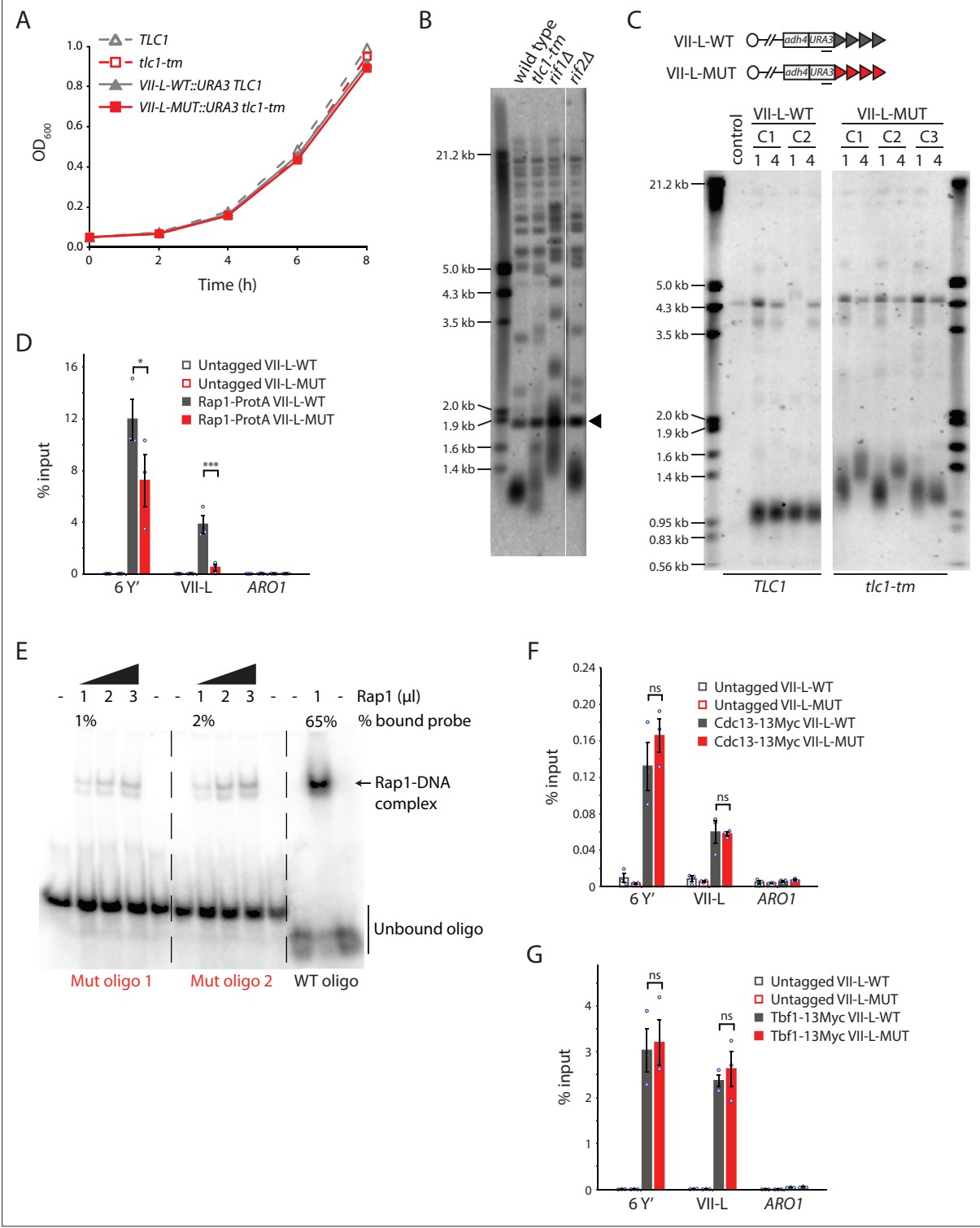

**Figure 1.** Rap1 binds poorly to *tlc1-tm* telomere sequences. (**A**) Exponentially growing cells of the indicated genotypes were diluted to an OD$_{600}$ of 0.05 and monitored for 8 hr. Three clones of each genotype were followed, and the average of the measurements at each time point is plotted. Error bars are too small to be visualized. (**B**) Telomere Southern blot analysis of strains of the indicated genotypes. Black arrowhead indicates a 1.8 kb DNA fragment generated from the BsmAI-digestion of plasmid pYt103 (**Shampay et al., 1984**). The major terminal restriction fragment is below the 1.8 kb control band. (**C**) Southern blot analysis of the artificial VII-L telomere, with either wild-type or *tlc1-tm* mutant sequence, using a probe to the adjacent *URA3* gene. Multiple clones were examined, with each clone propagated for 1–4 passages (each passage corresponds to approximately 25 generations). A wild-type strain (lacking the artificial VII-L telomere) was used as a control. (**D, F, G**) Chromatin immunoprecipitation coupled with quantitative PCR

*Figure 1 continued on next page*

*Figure 1 continued*

(ChIP-qPCR) analysis of the association of (**D**) protein A-tagged Rap1, (**F**) Myc-tagged Cdc13, and (**G**) Myc-tagged Tbf1 to six Y′ telomeres, the VII-L telomere, or to the non-telomeric *ARO1* locus. Untagged wild-type and *tlc1-tm* strains were used as controls. The mean percentage of input ± SEM is shown (n = 3, *p<0.05, ***p<10$^{-3}$). Source data are given in *Figure 1—source data 1*. Tbf1-binding motifs adjacent to the artificial VII-L telomere are shown in *Figure 1—figure supplement 1*. (**E**) Electrophoretic mobility shift assay (EMSA) of Rap1 protein incubated with radiolabeled oligonucleotides with either wild-type or *tlc1-tm* mutant telomeric sequence. The percentage of bound probe was determined by dividing the signal of the shifted band by the total signal (shifted plus unshifted). Uncropped blots for panels (**B**), (**C**), and (**E**) can be found in *Figure 1—source data 2*.

The online version of this article includes the following source data and figure supplement(s) for figure 1:

**Source data 1.** Chromatin immunoprecipitation.

**Source data 2.** Southern blots and EMSA.

**Figure supplement 1.** Tbf1-binding motifs adjacent to the artificial VII-L telomere.

assessed the association of Rap1 to the VII-L-WT or VII-L-MUT telomeres, as well as to native Y′ telomeres, by chromatin immunoprecipitation coupled with quantitative PCR (ChIP-qPCR). Binding of Rap1 is substantially reduced at the VII-L-MUT telomere (*Figure 1D*). Rap1 is more modestly reduced at Y′ telomeres due to the retention of some wild-type repeats at native telomeres. We confirmed the Rap1 ChIP-qPCR result by performing an electrophoretic mobility shift assay (EMSA) using recombinant full-length Rap1 protein. We find that Rap1 shows a very poor binding capacity to the *tlc1-tm* oligonucleotides compared to the wild-type telomeric sequence (*Figure 1E*).

Cdc13 is the main sequence-specific single-stranded telomeric DNA binding protein in *S. cerevisiae* (*Nugent et al., 1996*). Cdc13 requires a minimum of 11 nucleotides of GT-rich sequence for full binding affinity, with only three bases—forming a GNGT motif—recognized with high specificity (*Eldridge et al., 2006*). The GNGT motif is present in the *tlc1-tm* mutant sequence, so Cdc13 binding should not be altered. Consistent with this hypothesis, we find that Cdc13 associates equally well with both wild-type and mutant telomeres (*Figure 1F*).

The ability of *tlc1-tm* strains to survive with greatly reduced Rap1 telomere association was surprising given that Rap1 is the main telomere binding protein in *S. cerevisiae*, important for recruiting a number of other proteins to telomeres, and is encoded by an essential gene. However, the essential function of Rap1 is likely linked to its ability to bind nontelomeric sites in the genome where it acts as a transcriptional regulator of several hundred genes (*Azad and Tomar, 2016*). In addition, *S. cerevisiae* strains expressing a mutant telomerase that adds vertebrate TTAGGG telomeric repeats are devoid of Rap1, indicating that Rap1 is not essential for telomere capping (*Alexander and Zakian, 2003*; *Brevet et al., 2003*). In these strains, Tbf1 binds to the vertebrate repeats and is able to regulate telomere length homeostasis in a Rap1-independent manner (*Alexander and Zakian, 2003*; *Berthiau et al., 2006*). Thus, we assessed binding of Tbf1 to *tlc1-tm* telomeres. We find no change in the levels of Tbf1 at *tlc1-tm* telomeres compared to wild-type telomeres by ChIP-qPCR (*Figure 1G*), consistent with neither wild-type nor mutant telomere sequences containing the RCCCT Tbf1 consensus binding sequence (*Preti et al., 2010*), indicating that the loss of Rap1 is not compensated by recruitment of Tbf1. The Tbf1 association observed at the native Y′ telomeres is due to the presence of TTAGGG repeats at subtelomeric regions (*Brigati et al., 1993*), while the association observed at the artificial VII-L telomere is likely due to the presence of several RCCCT motifs in the adjacent sequence (*Figure 1—figure supplement 1*).

## Loss of Rap1-mediated telomere length regulation at *tlc1-tm* telomeres

The decreased binding of Rap1 to *tlc1-tm* sequences likely explains the long, heterogeneous-sized telomeres in *tlc1-tm* strains because Rap1 negatively regulates telomerase through what is called the 'protein counting' model. This model posits that Rap1, through its recruitment of Rif1 and Rif2, inhibits telomerase; the longer a telomere is, the more Rap1, Rif1, and Rif2 will be present, and the stronger the inhibition of telomerase will be (*Marcand et al., 1997*; *Levy and Blackburn, 2004*). Reduced binding of Rap1 would cause *tlc1-tm* telomeric sequences to not be recognized as telomeric in terms of Rap1-mediated telomere length regulation. To test this hypothesis, we again modified telomere VII-L to generate a telomere that would be seeded with either 84 bp or 300 bp of *tlc1-tm* telomeric sequence, but in a *TLC1* strain expressing wild-type telomerase so that the tip of the telomere would contain wild-type telomeric sequences. In both cases, the size of the telomere VII-L

terminal restriction fragment increased in size, with the magnitude of the increase roughly equivalent to the size of the *tlc1-tm* telomeric seed sequence, indicating that this sequence is not being sensed by the Rap1 protein counting mechanism (*Figure 2A*).

To examine telomere length regulation in *tlc1-tm* strains further, we monitored telomerase-mediated telomere extension events at nucleotide resolution after a single-cell cycle using the inducible Single Telomere EXtension (iSTEX) assay (*Strecker et al., 2017*). At wild-type telomeres, telomerase extends only a subset of telomeres in each cell cycle, with a strong preference for the extension of short telomeres (*Teixeira et al., 2004*; *Strecker et al., 2017*). At *tlc1-tm* telomeres, we find that telomere extension frequency is dramatically increased, with nearly all (92%) of the telomeres being extended during a single-cell cycle (*Figure 2B and C*). This observation is consistent with the protein counting model; the depletion of Rap1 at *tlc1-tm* telomeres causes them to all be recognized as short telomeres in need of elongation.

## *tlc1-tm* telomeres are rapidly degraded in the absence of telomerase

The increase in telomere extension frequency of *tlc1-tm* telomeres is much more dramatic than the increase seen in *rif1Δ*, *rif2Δ*, and *pif1-m2* cells (*Teixeira et al., 2004*; *Stinus et al., 2017*), even though these mutants have similar or longer telomeres than the *tlc1-tm* mutant (e.g., see *Figure 1B* for comparison to *rif1Δ* and *rif2Δ*). Ablation of Rap1 leads to readily detectable telomere degradation within a few hours (*Vodenicharov et al., 2010*), so we hypothesized that while decreased Rap1 at *tlc1-tm* telomeres may favor their extension by telomerase, increased degradation may limit the length of *tlc1-tm* telomeres. If true, removal of telomerase should trigger rapid entry into senescence. To test this idea, we sporulated diploid strains that were heterozygous for *EST2*, which encodes the protein catalytic subunit of telomerase (*Lingner et al., 1997*), or *TLC1*, with either wild-type or mutant telomeres (i.e., *est2Δ/EST2* versus *est2Δ/EST2 tlc1-tm/tlc1-tm* and *tlc1Δ/TLC1* versus *tlc1Δ/tlc1-tm*), and performed senescence assays with the haploid meiotic progeny. In the presence of wild-type telomeres but absence of telomerase (*est2Δ* and *tlc1Δ*), telomeres shortened until the cells senesced after 60–70 population doublings, as expected (*Figure 3A*). A small subset of the senescent population was then able to lengthen the telomeres by recombination-mediated mechanisms, forming so-called survivors (*Lundblad and Blackburn, 1993*). We found that cells containing mutant telomeres senesced rapidly, only ~40 population doublings after telomerase loss (*Figure 3A*, *est2Δ tlc1-tm\** and *tlc1Δ\**). We examined the telomere length of cells that had undergone ~30 population doublings after isolation of the haploid spores and found that, upon loss of telomerase, mutant telomeres became extremely heterogeneous and degraded (i.e., a continuous smear of telomeric DNA-hybridizing signal extending from the wells to the bottom; *Figure 3B*).

## Rif1 and Sir4, but not Rif2 nor Sir3, association to *tlc1-tm* telomeres is decreased

Rap1, through its C-terminal domain, recruits Rif1, Rif2, Sir3, and Sir4 to telomeres (*Hardy et al., 1992*; *Wotton and Shore, 1997*; *Moretti et al., 1994*). We reasoned that the reduction of Rap1 at the *tlc1-tm* telomeres should also decrease recruitment of Rif1, Rif2, Sir3, and Sir4, which we tested by ChIP-qPCR (*Figure 4A*). Consistent with this hypothesis, we find that Rif1 and Sir4 recruitment to the VII-L-MUT telomere is reduced compared to the VII-L-WT telomere (to 16% for Rif1 and 31% for Sir4). Surprisingly, the recruitment of neither Rif2 nor Sir3 is affected. Rif2 can interact with the Xrs2 and Rad50 subunits of the MRX complex (*Hirano et al., 2009*; *Hailemariam et al., 2019*; *Roisné-Hamelin et al., 2021*), which binds to DNA ends and telomeres (*Oh and Symington, 2018*); Rif2 can also interact directly with double-strand DNA (*Cassani et al., 2016*; *Hailemariam et al., 2019*). Sir3 possesses multiple domains that can interact with histones (*Gartenberg and Smith, 2016*). Thus, Rif2 and Sir3 may associate to *tlc1-tm* telomeres via Rap1-independent mechanisms.

## Transcription is not affected at *tlc1-tm* telomeres

Rap1, through its recruitment of the Rif and Sir proteins, represses the transcription of non-coding telomeric repeat-containing RNA (TERRA) (*Iglesias et al., 2011*). Reduced levels of Rap1, Rif1, and Sir4 at *tlc1-tm* telomeres may alter TERRA transcription. However, we find no change in TERRA abundance in *tlc1-tm* cells (*Figure 4B*). Rap1, through its recruitment of the Sir complex, also mediates the transcriptional silencing of subtelomeric genes (*Moretti et al., 1994*). The VII-L-WT and VII-L-MUT

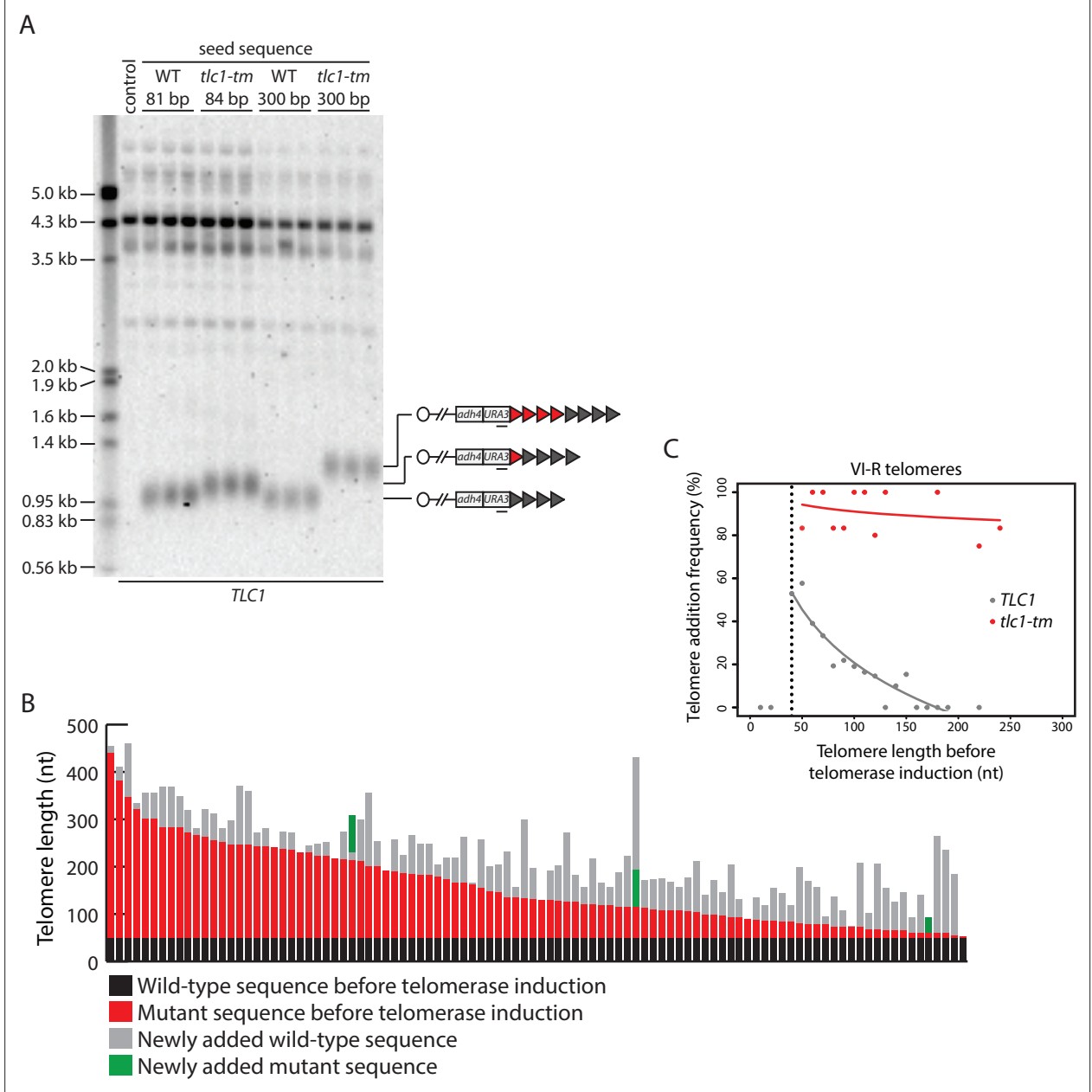

**Figure 2.** Telomere length regulation is disrupted in *tlc1-tm* cells. (**A**) Southern blot analysis of the artificial VII-L telomere using a probe to the adjacent *URA3* gene. The telomere was seeded with either wild-type or *tlc1-tm* mutant sequence of the indicated lengths in a strain expressing wild-type TLC1. Multiple clones of each strain were examined. A wild-type strain (lacking the artificial VII-L telomere) was used as a control. The uncropped blot can be found in *Figure 2—source data 1*. (**B**) In vivo extension of *tlc1-tm* telomeres was examined using the inducible Single Telomere EXtension (iSTEX) assay. Telomere VI-R was amplified and sequenced after the induction of wild-type telomerase. Each bar represents an individual telomere. The black and red portions of each bar represent wild-type and *tlc1-tm* sequence, respectively, that is identical in sequence and thus present before telomerase induction. The length of the black/wild-type sequence is 48 bp. Sequence that is divergent from the black and red sequence is shown in gray and green. Gray represents newly added wild-type sequence after the induction of telomerase. Green represents divergent *tlc1-tm* sequence, most likely a result of homologous recombination. Telomeres are sorted based on the length of the undiverged (black plus red) sequence. (**C**) Telomere VI-R sequences obtained from the iSTEX analysis in (**B**) were binned into groups of 10 nt in size according to telomere length before telomerase induction. iSTEX data for the extension of wild-type telomeres were taken from previous studies (*Strecker et al., 2017*; *Stinus et al., 2017*) and included for comparison. Groups containing less than four telomeres were excluded from this analysis. Frequency of extension and average telomere length before telomerase induction were calculated and plotted for each group. Logarithmic regression curves for each dataset were also included in the plot. Telomeres shorter than 40 nt before telomerase induction, which are not efficiently recognized and extended by telomerase (*Strecker et al., 2017*), were removed from the regression analysis. Source data are given in *Figure 2—source data 2*.

*Figure 2 continued on next page*

*Figure 2 continued*

The online version of this article includes the following source data for figure 2:

**Source data 1.** Southern blots.

**Source data 2.** iSTEX assay.

telomeres possess a *URA3* gene immediately adjacent to the telomeric repeats. Silencing of the *URA3* gene was monitored by assaying for growth in the presence of 5-fluoroorotic acid (5-FOA), which kills cells expressing *URA3* (*Figure 4C*). Strains deleted for *SIR2*, which causes a defect in silencing (*Aparicio et al., 1991*), were used as controls. Because hypersensitivity to 5-FOA does not correlate with loss of silencing in some genetic backgrounds (*Rossmann et al., 2011*), we also measured URA3 transcript levels by reverse-transcriptase quantitative PCR (*Figure 4D*). In both assays, we find no difference between VII-L-WT and VII-L-MUT, indicating that the reduced levels of Rap1 and Sir4 at *tlc1-tm* telomeres still allow functional silencing.

## Rif2 prevents degradation of *tlc1-tm* telomeres

Rap1 regulates telomere length homeostasis through its recruitment of Rif1, Rif2, and Sir4. Rif1 and Rif2 negatively regulate telomerase (*Hardy et al., 1992*; *Wotton and Shore, 1997*; *Levy and Blackburn, 2004*). In contrast, Sir4 functions in a pathway to recruit telomerase to telomeres (*Hass and Zappulla, 2015*). We tested what effect deleting *RIF1*, *RIF2*, and *SIR4* would have on telomere length in *tlc1-tm* cells. Deletion of *SIR4* results in a small decrease in telomere length in *TLC1* cells, as expected (*Hass and Zappulla, 2015*), but no noticeable change in *tlc1-tm* cells (*Figure 5A*). The lack of an effect could be due to Sir4 already being reduced at *tlc1-tm* telomeres (*Figure 4A*). Deletion of

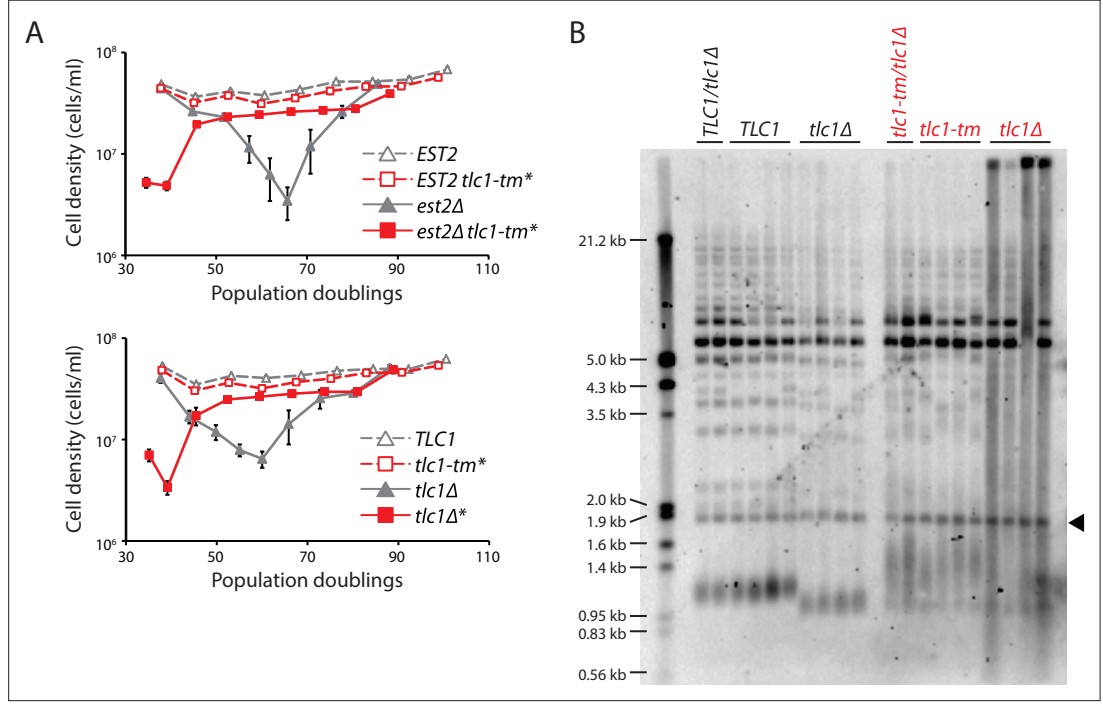

**Figure 3.** Fast degradation of *tlc1-tm* telomeres in the absence of telomerase. (**A**) Senescence rates were measured by serial passaging of strains of the indicated genotypes, derived from the sporulation of *est2Δ/EST2* and *est2Δ/EST2 tlc1-tm/tlc1-tm* diploids (top panel) or *tlc1Δ/TLC1* and *tlc1Δ/tlc1Δ/tlc1-tm* diploids (bottom panel). Cell density was measured every 24 hr, followed by dilution to 1 × 10⁵ cells/ml. Mean ± SEM of four independent isolates per genotype is plotted. (**B**) Telomere Southern blot analysis of samples obtained at the first time point of the senescence assays in (**A**). Black arrowhead indicates the 1.8 kb telomere sequence-containing fragment loaded as control, as in *Figure 1B*. The uncropped blot can be found in *Figure 3—source data 1*.

The online version of this article includes the following source data for figure 3:

**Source data 1.** Southern blot.

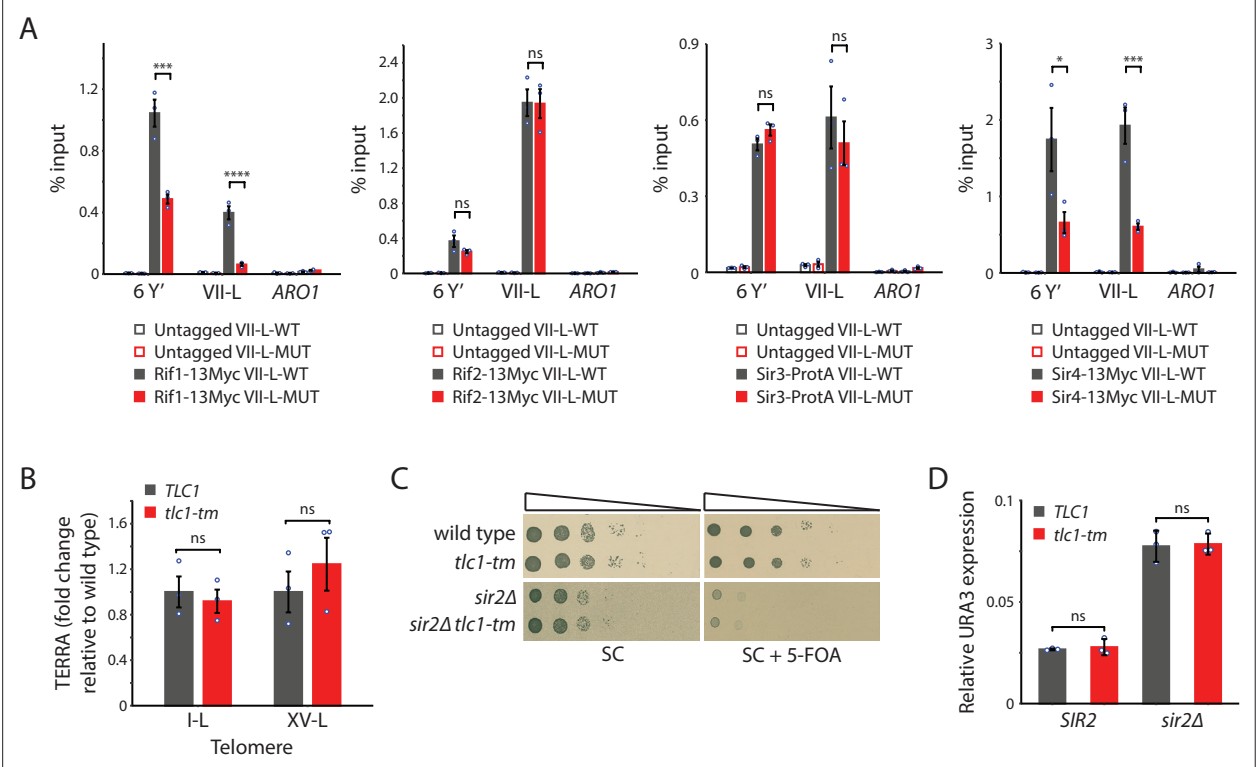

**Figure 4.** Recruitment of Rif1 and Sir4, but not Rif2 nor Sir3, is significantly reduced at *tlc1-tm* telomeres. (**A**) Chromatin immunoprecipitation coupled with quantitative PCR (ChIP-qPCR) analysis of the association of Myc-tagged Rif1, Rif2, Sir4, and protein A-tagged Sir3 to six Y′ telomeres, the VII-L telomere, or to the non-telomeric *ARO1* locus. Untagged wild-type and *tlc1-tm* strains were used as controls. (**B**) Total RNA was reverse transcribed and telomeric repeat-containing RNA (TERRA) from specific telomeres (I-L and XV-L) was analyzed by qPCR. TERRA values were normalized to *ACT1* levels, and to the respective wild type (*TLC1*). (**C**) Tenfold serial dilutions of strains with the indicated genotypes were spotted on SC plates without or with 5-fluoroorotic acid (5-FOA). (**D**) The expression of the subtelomerically integrated *URA3* gene was measured in the indicated yeast strains by RT-qPCR. All data (except **C**) are shown as mean ± SEM (n = 3, ***p<10⁻³, ****p<10⁻⁴). Source data are given in *Figure 4—source data 1*.

The online version of this article includes the following source data and figure supplement(s) for figure 4:

**Source data 1.** Chromatin immunoprecipitation.

**Source data 2.** Southern blot.

**Figure supplement 1.** Examining the effect of deleting *SIR3* in *tlc1-tm* and *tlc1-476A* cells.

*RIF1* increases telomere length in both *TLC1* and *tlc1-tm* cells (*Figure 5B*, *Figure 5—figure supplement 1*), despite Rif1 also being reduced at *tlc1-tm* telomeres (*Figure 4A*). Rif1 may still impact *tlc1-tm* telomeres because it plays a much bigger role in telomere length homeostasis than Sir4.

Deletion of *RIF2* causes dramatic degradation of *tlc1-tm* telomeres, with a continuous smear of telomeric DNA-hybridizing signal observed (*Figure 5B*, *Figure 5—figure supplement 1*), similar to *tlc1-tm* cells upon loss of telomerase (*Figure 3B*). This finding, along with the observation that Rif2 is still present at *tlc1-tm* telomeres (*Figure 4A*), indicates that Rif2 has an important role in protecting *tlc1-tm* telomeres. Since uncapped telomeres expose chromosome ends to non-homologous end joining (NHEJ) and homologous recombination (HR), we tested whether the *rif2Δ tlc1-tm* telomere profile would change by deleting *DNL4* or *RAD52*, which are required for NHEJ and HR, respectively (*Wilson et al., 1997*; *Symington et al., 2014*). The extensive smear was still observed in *dnl4Δ rif2Δ tlc1-tm* and *rad52Δ rif2Δ tlc1-tm* triple mutants, but was reduced in intensity, especially at low molecular weight, in *rad52Δ rif2Δ tlc1-tm* (*Figure 5C*). *rif2Δ tlc1-tm* cells grow poorly, and this growth defect is exacerbated by additional deletion of *RAD52*; *rad52Δ rif2Δ tlc1-tm* spores mostly fail to germinate, and those that do grow even more poorly than *rif2Δ tlc1-tm* (*Figure 5D and E*), suggesting that Rad52-mediated HR helps cope with *rif2Δ tlc1-tm* uncapped telomeres. *rad52Δ tlc1-tm* cells also

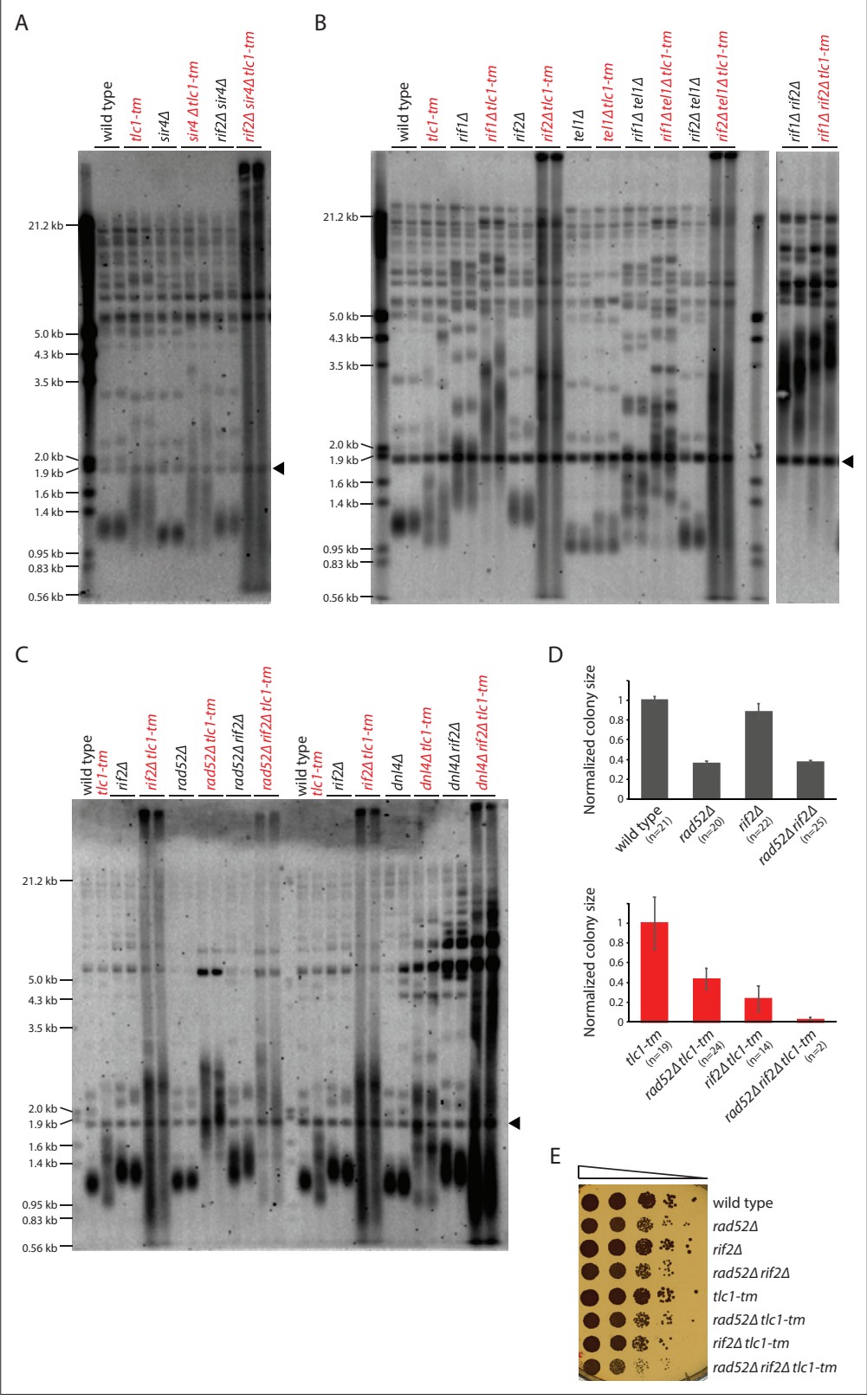

**Figure 5.** Rif2 prevents degradation of *tlc1-tm* telomeres. (**A–C**) Telomere Southern blot analysis of strains of the indicated genotypes. Black arrowhead indicates the 1.8 kb telomere sequence-containing fragment loaded as control, as in ***Figure 1B***. A Southern blot analysis of the artificial VII-L telomere using a probe to the adjacent *URA3* gene in strains from (**B**) is shown in ***Figure 5—figure supplement 1***. The slight differences in the telomere profile

*Figure 5 continued on next page*

*Figure 5 continued*

of isogenic *tlc1-tm* strains are examined further in *Figure 5—figure supplement 2*. The uncropped blots can be found in *Figure 5—source data 1*. (**D**) Colony sizes of haploid meiotic progeny derived from the sporulation of *rad52Δ/RAD52 rif2Δ/RIF2* and *rad52Δ/RAD52 rif2Δ/RIF2 tlc1-tm/tlc1-tm* diploid strains were measured and normalized to wild-type (top panel) or *tlc1-tm* (bottom panel). The number of colonies analyzed is indicated in parenthesis. Error bars show SEM. (**E**) Tenfold serial dilutions of strains with the indicated genotypes were spotted on a YPD plate and grown at 30°C.

The online version of this article includes the following source data and figure supplement(s) for figure 5:

**Source data 1.** Southern blot.

**Figure supplement 1.** Rif2 protects *tlc1-tm* telomeres from degradation.

**Figure supplement 2.** Slight differences in the telomere profile of isogenic *tlc1-tm* strains are not due to Rad52-dependent homologous recombination.

---

have longer telomeres than *tlc1-tm* cells (*Figure 5C*), indicating that HR is important for maintaining *tlc1-tm* telomeres.

Interestingly, *rif1Δ rif2Δ* and *rif1Δ rif2Δ tlc1-tm* cells have very similar telomere profiles: very long telomeres but without the extensive degradation seen in *rif2Δ tlc1-tm* cells (*Figure 5B*). This observation is reminiscent of the elevated levels of telomere-telomere fusion seen in *rif2Δ tel1Δ* cells, and not in *rif1Δ rif2Δ tel1Δ* cells (*Marcand et al., 2008*), but it is currently unclear whether there is a common underlying mechanism that explains both observations.

## Rif2 protects *tlc1-tm* telomeres by inhibiting the MRX complex

*TEL1* is epistatic to *RIF2* with respect to telomere length regulation; while *rif2Δ* cells have long telomeres, *tel1Δ* and *rif2Δ tel1Δ* cells have very short telomeres (*Craven and Petes, 1999*). Thus, we tested whether the same would be true in terms of the telomere degradation seen in *rif2Δ tlc1-tm* cells. Deletion of *TEL1* shortens the telomeres of *tlc1-tm* cells, but does not affect the telomere profile of *rif2Δ tlc1-tm* cells (*Figure 5B*), indicating that Rif2 does not inhibit Tel1 to prevent the degradation of *tlc1-tm* telomeres. Recent studies have shown that Rif2 attenuates Tel1 activity at telomeres by inhibiting the MRX complex (consisting of Mre11, Rad50, and Xrs2), which is responsible for recruiting Tel1 to telomeres (*Hailemariam et al., 2019*; *Sabourin et al., 2007*). Rif2 discharges the ATP-bound state of Rad50, thereby making the MRX complex incapable of activating Tel1 (*Hailemariam et al., 2019*). Similarly, Rif2 enhancement of Rad50 ATPase activity limits MRX-mediated tethering of DNA ends during double-strand break (DSB) repair (*Cassani et al., 2016*). Rif2 also inhibits MRX-mediated resection of telomeric DNA ends (*Bonetti et al., 2010a*; *Bonetti et al., 2010b*). Rif2 inhibition of the MRX complex involves a direct interaction between the BAT/MIN motif of Rif2 with the ATPase domain of Rad50 (*Roisné-Hamelin et al., 2021*; *Khayat et al., 2021*). Therefore, we hypothesized that Rif2 could be inhibiting MRX-mediated degradation of *tlc1-tm* telomeres. Consistent with this idea, we find that Mre11 is about fivefold more associated with the VII-L-MUT telomere in comparison with the VII-L-WT telomere (*Figure 6A*), and that the telomere degradation observed in *rif2Δ tlc1-tm* cells is absent in *rad50Δ rif2Δ tlc1-tm* cells (*Figure 6B*).

The *rad50Δ tlc1-tm* and *rad50Δ rif2Δ tlc1-tm* cells (also *cdc13-1 tlc1-tm* cells, discussed below) show a telomere profile reminiscent of type II survivors, which maintain their telomeres in a telomerase-independent manner that relies on recombination-mediated amplification of telomeric sequence (*Lundblad and Blackburn, 1993*; *Teng and Zakian, 1999*). Independently generated isogenic isolates of type II survivors will have similar but slightly different telomere profiles, much like that observed in different isolates of isogenic *tlc1-tm* strains (as seen in *rad50Δ tlc1-tm* and *rad50Δ rif2Δ tlc1-tm* cells, but also in *tlc1-tm* strains in general). However, Rad50 is required for type II survivor formation (*Chen et al., 2001*), and this effect is still observed in *rad50Δ tlc1-tm* and *rad50Δ rif2Δ tlc1-tm* strains with an additional deletion of *RAD52*, which is required for HR and survivor formation (*Lundblad and Blackburn, 1993*; *Claussin and Chang, 2015*; *Figure 5—figure supplement 2A*). Interestingly, the severe growth defect of *rad52Δ rif2Δ tlc1-tm* strains (*Figure 5D*) is suppressed by an additional deletion of *RAD50* (*Figure 5—figure supplement 2B*). At present, we do not know the reason for the slight telomere profile differences seen between different isolates of isogenic *tlc1-tm* strains, but it may be related to a previous observation that the very tip of the telomere has unique sequence features that affect telomere capping and length regulation (*Grossi et al., 2001*).

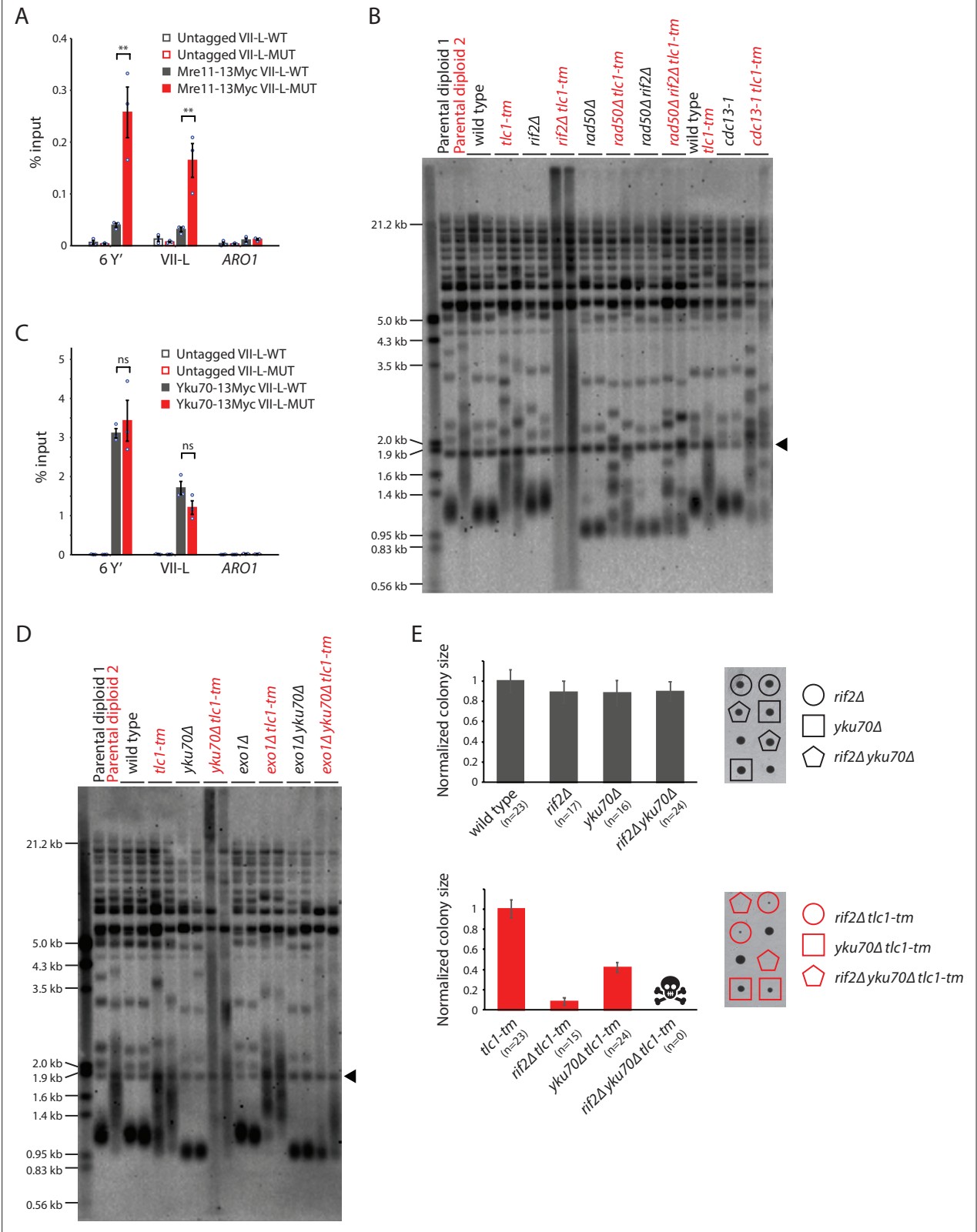

**Figure 6.** Rif2 protects *tlc1-tm* telomeres by inhibiting the MRX complex and acts in parallel with the Yku complex. (**A, C**) Chromatin immunoprecipitation coupled with quantitative PCR (ChIP-qPCR) analysis of the association of Myc-tagged Mre11 and Yku70 to six Y′ telomeres, the VII-L telomere, or to the non-telomeric *ARO1* locus. Untagged wild-type and *tlc1-tm* strains were used as controls. The mean percentage of input ± SEM is shown (n = 3, **p<0.01). Source data are given in *Figure 6—source data 1*. (**B, D**) Telomere Southern blot analysis of strains of the indicated

*Figure 6 continued on next page*

*Figure 6 continued*

genotypes. Black arrowhead indicates the 1.8 kb telomere sequence-containing fragment loaded as control, as in *Figure 1B*. (**B**) Parental diploid 1 is *rad50Δ/RAD50 rif2Δ/RIF2*; parental diploid 2 is *rad50Δ/RAD50 rif2Δ/RIF2 tlc1-tm/tlc1-tm*. The last six lanes are derived from cells grown at 22°C. (**D**) Parental diploid 1 is *exo1Δ/EXO1 yku70Δ/YKU70*; parental diploid 2 is *exo1Δ/EXO1 yku70Δ/YKU70 tlc1-tm/tlc1-tm*. The uncropped blots can be found in *Figure 6—source data 2*. (**E**) Colony sizes of haploid meiotic progeny derived from the sporulation of *rif2Δ/RIF2 yku70Δ/YKU70* and *rif2Δ/RIF2 yku70Δ/YKU70 tlc1-tm/tlc1-tm* diploid strains were measured and normalized to wild-type (top panel) or *tlc1-tm* (bottom panel). The number of colonies analyzed is indicated in parenthesis. Error bars show SEM. Representative images of dissected tetrads are shown on the right. Each column of colonies arose from a single tetrad.

The online version of this article includes the following source data and figure supplement(s) for figure 6:

**Source data 1.** Chromatin immunoprecipitation.

**Source data 2.** Southern blots.

**Figure supplement 1.** Examining the effect of deleting *RAD52* in *cdc13-1 tlc1-tm*.

## Rif2 and the Yku complex act in parallel to protect *tlc1-tm* telomeres

The CST (Cdc13-Stn1-Ten1) and Yku (Yku70-Yku80) complexes are also important for inhibiting nucleolytic degradation at telomeres (*Garvik et al., 1995*; *Grandin et al., 1997*; *Grandin et al., 2001*; *Gravel et al., 1998*), and their importance might be increased at *tlc1-tm* telomeres. Cdc13 and Yku70 are present at similar levels at VII-L-WT and VII-L-MUT telomeres (*Figure 1F*, *Figure 6C*). The *cdc13-1* temperature-sensitive mutant causes an accumulation of single-stranded telomeric DNA (*Garvik et al., 1995*), and is defective even at the permissive temperature of 23°C (*Paschini et al., 2012*). *cdc13-1 tlc1-tm* cells do not exhibit the extensive telomere degradation seen in *rif2Δ tlc1-tm* cells (*Figure 6B*). Independently generated *cdc13-1 tlc1-tm* cells show slightly different telomere profiles. However, unlike *rad50Δ tlc1-tm* and *rad50Δ rif2Δ tlc1-tm* cells, this heterogeneity disappears upon deletion of *RAD52* (*Figure 6—figure supplement 1A*). *cdc13-1 rad52Δ tlc1-tm* triple mutants grow very poorly (*Figure 6—figure supplement 1B*), suggesting that Rad52-mediated HR plays a role in maintaining *cdc13-1 tlc1-tm* telomeres. In contrast to *cdc13-1 tlc1-tm* cells, we observe extensive telomere degradation in *yku70Δ tlc1-tm* cells (*Figure 6D*). Telomere degradation is not seen in *exo1Δ yku70Δ tlc1-tm* cells, suggesting that the Yku complex inhibits the Exo1 exonuclease to protect *tlc1-tm* telomeres. *rif2Δ yku70Δ tlc1-tm* cells are not viable (*Figure 6E*), indicating that Rif2 and the Yku complex function in parallel to protect *tlc1-tm* telomeres.

## Rif2 recruitment to *tlc1-tm* telomeres requires a DNA end and the MRX complex

The telomere degradation seen in *rif2Δ tlc1-tm* cells has also been observed in other *tlc1* template mutants, such as the *tlc1-476A* mutant (*Figure 7A*), even without deletion of *RIF2* (*Chan et al., 2001*; *Lin et al., 2004*). The *tlc1-476A* mutant changes the invariable CCC core of the template region to CAC, causing the addition of long stretches of TG dinucleotide repeats, occasionally interrupted by a TGG trinucleotide, that are predicted to disrupt Rap1 binding even more so than *tlc1-tm* repeats (*Chan et al., 2001*; *Graham and Chambers, 1994*). We find that *tlc1-476A* mutants are completely dependent on Rif2 for survival as *tlc1-476A rif2Δ* double mutants are inviable (*Figure 7B*).

To examine how Rif2 is able to associate with the Rap1-depleted telomeres of *tlc1-tm* cells, we first asked whether a DNA end is required. We inserted 300 bp of either wild-type or *tlc1-tm* telomeric sequence at an internal genomic locus on chromosome III (III-ITS; chromosome III interstitial telomeric sequence) and assessed Rif2 association by ChIP-qPCR. We find that Rif2 associates with the wild-type ITS, but not the *tlc1-tm* ITS (*Figure 7C*), indicating that Rap1-independent recruitment of Rif2 to *tlc1-tm* sequence requires a DNA end. Since the MRX complex binds to DNA ends and telomeres (*Oh and Symington, 2018*), and Rif2 is known to interact with both Rad50 and Xrs2 (*Hirano et al., 2009*; *Hailemariam et al., 2019*; *Roisné-Hamelin et al., 2021*), Rif2 recruitment to *tlc1-tm* telomeres might require the MRX complex. Consistent with this hypothesis, we find that Rif2 association to the *tlc1-tm* telomeres is lost in *rad50Δ* strains (*Figure 7D*).

Since Rap1 is not completely absent at *tlc1-tm* telomeres, we asked whether the Rap1-Rif2 interaction was still important for protecting *tlc1-tm* telomeres. We transformed *rif2Δ* and *rif2Δ tlc1-tm* cells with a centromeric plasmid expressing Rif2 or a Rif2-L44R,V45E,E347R mutant that is unable to interact with Rap1 (*Shi et al., 2013*). The cells were then passaged for approximately 100 generations,

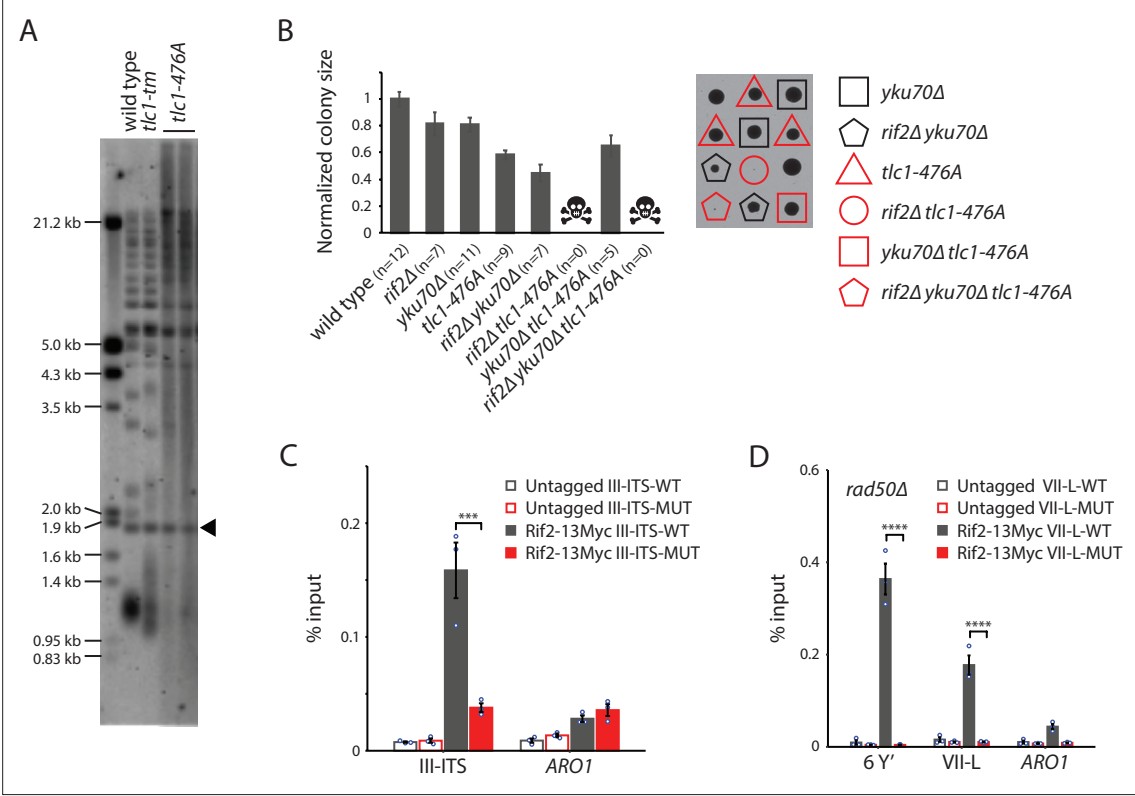

**Figure 7.** Rif2 association to *tlc1-tm* telomeres requires the MRX complex. (**A**) Telomere Southern blot analysis of strains of the indicated genotypes. Black arrowhead indicates the 1.8 kb telomere sequence-containing fragment loaded as control, as in *Figure 1B*. The uncropped blot can be found in *Figure 7—source data 1*. (**B**) Colony sizes of haploid meiotic progeny derived from the sporulation of a *rif2Δ/RIF2 yku70Δ/YKU70 tlc1-476A/TLC1* diploid strain were measured and normalized to wild type. The number of colonies analyzed is indicated in parenthesis. Error bars show SEM. Representative images of dissected tetrads are shown on the right. Each column of colonies arose from a single tetrad. (**C**) Chromatin immunoprecipitation coupled with quantitative PCR (ChIP-qPCR) analysis of the association of Myc-tagged Rif2 to a 300 bp ITS located on chromosome III containing either wild-type or *tlc1-tm* sequence. The mean percentage of input ± SEM is shown (n = 3, ***p<10⁻³). (**D**) ChIP-qPCR analysis of the association of Myc-tagged Rif2 to six Y' telomeres, the VII-L telomere, or to the non-telomeric *ARO1* locus in a *rad50Δ* background. Untagged strains were used as controls. The mean percentage of input ± SEM is shown (n = 3, ****p<10⁻⁴). Source data for ChIP experiments are given in *Figure 7—source data 2*.

The online version of this article includes the following source data and figure supplement(s) for figure 7:

**Source data 1.** Southern blots.

**Source data 2.** Chromatin immunoprecipitation.

**Figure supplement 1.** Expression of a Rif2-L44R,V45E,E347R mutant suppresses neither the telomere degradation nor slow growth of *rif2Δ tlc1-tm* cells.

which was sufficient for Rif2, but not Rif2-L44R,V45E,347R, expression to partially restore the telomere profile and suppress the growth defect of *rif2Δ tlc1-tm* cells (*Figure 7—figure supplement 1*). Thus, the Rap1-Rif2 interaction is still important in *tlc1-tm* cells. Alternatively, the L44R, V45E, and E347R mutations may disrupt Rif2 protection of *tlc1-tm* telomeres in a manner independent of its ability to interact with Rap1.

## Discussion

In this study, we investigated the consequences of depleting Rap1 from *S. cerevisiae* telomeres. Our findings suggest that impairing Rap1 binding does not significantly affect cell proliferation. The depletion of telomeric Rap1 causes defects in telomere length regulation and telomere capping. Surprisingly, Rif2 is still recruited to Rap1-depleted telomeres, and its recruitment is dependent on the MRX complex. *tlc1-tm* telomeres become extensively degraded in the absence of Rif2 or the Yku complex, and *rif2Δ yku70Δ tlc1-tm* triple mutants are inviable.

Telomere capping is essential for cell viability; yeast cells cannot survive the loss of a single telomere (**Sandell and Zakian, 1993**). Telomere capping is executed by telomere-binding proteins, so it was surprising to find that yeast cells can survive when Rap1, the main protein that binds double-stranded telomeric DNA, is significantly reduced at telomeres. Previous studies have reported that telomeres of *S. cerevisiae* expressing mutant telomerase that adds TTAGGG repeats are devoid of Rap1; however, the absence of Rap1 in this situation appears to be compensated by the binding of Tbf1 to the TTAGGG sequence (**Alexander and Zakian, 2003**; **Brevet et al., 2003**; **Berthiau et al., 2006**). In contrast, there is no change in Tbf1 binding at *tlc1-tm* telomeres (**Figure 1G**). The difference between yeast telomeres with TTAGGG repeats and *tlc1-tm* telomeres is further highlighted by the fact that Rif2 is absent at the former (**Alexander and Zakian, 2003**), but is crucial for telomere protection of the latter. In addition, multiple viable *tlc1* template mutants, including *tlc1-tm*, result in the addition of mutant telomeric sequence that does not conform to the consensus binding sites of both Rap1 and Tbf1 (**Prescott and Blackburn, 2000**; **Förstemann et al., 2003**; **Lin et al., 2004**), supporting the notion that telomeres can remain sufficiently capped in the absence of both proteins.

Previously studied *tlc1* template mutations that abolish Rap1 binding to the resulting mutant telomere sequences generally fall into two categories; they either cause telomere shortening or they cause rapid telomere elongation, which in some cases is accompanied by extensive telomere degradation (**Prescott and Blackburn, 2000**; **Lin et al., 2004**), similar to that seen in *rif2Δ tlc1-tm* mutants (**Figure 5B**). The first category is likely due to a loss of telomerase enzymatic activity because most *tlc1* template mutations result in a reduction in the nucleotide addition processivity of telomerase (**Förstemann et al., 2003**). The extent of telomere elongation and degradation in the second category appears to be correlated with the decrease in Rap1 binding (**Prescott and Blackburn, 2000**). Rap1 association to fully mutant *tlc1-tm* telomeres is reduced to approximately 13% compared to wild-type telomeres (**Figure 1D**). We suspect that this level of Rap1 prevents the more extensive elongation and degradation seen in several other *tlc1* template mutants (e.g., *tlc1-476A*; **Figure 7A**) that likely have even less Rap1 telomere association (**Prescott and Blackburn, 2000**; **Lin et al., 2004**). Consistent with this hypothesis, *tlc1Δ* cells derived from the sporulation of *tlc1-tm/tlc1Δ* diploids with mutant telomeres also exhibit extensive degradation (**Figure 3B**), which is likely the result of further reducing Rap1 telomere association due to telomere shortening.

Our findings build upon previous work to show that Rap1 and Rif2 inhibit the MRX complex to prevent telomere degradation. First, decreased telomere association of Rap1 in *tlc1-tm* cells (**Figure 1D**) is accompanied by an increase in Mre11 telomere association (**Figure 6A**), which is consistent with a previous report showing that Rap1 binding inhibits Mre11 recruitment (**Negrini et al., 2007**). Second, the telomere degradation observed in *rif2Δ tlc1-tm* cells is eliminated by deletion of *RAD50* (**Figure 6B**), which is consistent with previous studies showing that Rif2 inhibits the MRX complex by discharging the ATP-bound form of Rad50 (**Cassani et al., 2016**; **Hailemariam et al., 2019**). Rif2 and the Yku complex have previously been shown to separately inhibit the resection of telomeric ends in *TLC1* cells, but *rif2Δ ykuΔ* double mutants remain viable (**Bonetti et al., 2010a**; **Bonetti et al., 2010b**). In contrast, we find that *rif2Δ yku70Δ tlc1-tm* cells are inviable (**Figure 6E**), indicating that Rap1 has an important role alongside Rif2 and the Yku complex to inhibit telomere degradation. Furthermore, the inviability of *tlc1-476A rif2Δ* double mutants (**Figure 7B**) indicates that Rap1 and Rif2 are more important than the Yku complex in this regard.

The redundant capping mechanisms offer an explanation for the rapid evolution of the telomere sequence and telomere-binding proteins in budding yeast species of the Saccharomycotina subdivision, of which *S. cerevisiae* is a member (**Steinberg-Neifach and Lue, 2015**). Alterations in the template region of telomerase RNA or in the DNA-binding domain of telomere-binding proteins are more easily tolerated if there are redundant mechanisms to ensure sufficient telomere capping and, therefore, cell viability. Indeed, the budding yeast *Kluyveromyces lactis* can also tolerate changes to the template region of its telomerase RNA that disrupts binding to Rap1 (**Krauskopf and Blackburn, 1996**). The more distantly related fission yeast *Schizosaccharomyces pombe* can also survive without its main telomere-binding protein, Taz1 (**Cooper et al., 1997**), suggesting that redundancy of telomere capping mechanisms has also evolved independently outside of the Saccharomycotina subdivision.

The association of Rif2 to Rap1-depleted *tlc1-tm* telomeres was unexpected given the well-characterized role of the Rap1 C-terminal domain in recruiting Rif2 to telomeres (**Wotton and Shore,**

*1997*; *Shi et al., 2013*). We find that this Rap1-independent recruitment of Rif2 is dependent on the MRX complex (*Figure 7D*), which itself is increased at *tlc1-tm* telomeres (as determined by ChIP-qPCR analysis of Mre11; *Figure 6A*). The MRX complex plays a central role in the repair of DSBs (*Oh and Symington, 2018*), so why does Rif2 play such a negligible role? It has recently been reported that Rif2 and Sae2 have opposing roles in regulating the MRX complex, and the role of Rif2 in DSB repair is more readily detected in the absence of Sae2 (*Marsella et al., 2021*). While Sae2 has a prominent role in DSB repair, it has a minor role at telomeres; the inverse is true of Rif2 at DSBs and telomeres (*Bonetti et al., 2021*). Thus, Sae2 appears to limit Rif2 function at DSBs, and Rif2 may limit the role of Sae2 at telomeres. In wild-type cells, the prominent role of Rif2 at telomeres can be explained by its recruitment by Rap1. However, it is unclear how Rif2 maintains its prominent role at telomeres without Rap1-dependent recruitment. One possibility is that there is preferential recruitment of Sae2 to DSBs compared to telomeres. Further studies are needed to examine this hypothesis.

Like Rif2, Sir3 is normally recruited to telomeres via the C-terminal domain of Rap1 (*Moretti et al., 1994*), so it was also unexpected to still find Sir3 associated to Rap1-depleted *tlc1-tm* telomeres and subtelomeric gene silencing intact (*Figure 4*). Sir3 was recently reported to inhibit DSB resection via two pathways, one dependent on the assembly of heterochromatin and the other involving direct physical interaction and inhibition of Sae2 (*Bordelet et al., 2021*). Deletion of *SIR3* in *tlc1-tm* cells increased telomere length, but did not cause telomere degradation seen in *rif2Δ tlc1-tm* cells, and did not affect cell growth (*Figure 4—figure supplement 1*). Deletion of *SIR3* in *tlc1-476A* cells affects neither telomere profile nor cell growth. Thus, the presence of Sir3 at *tlc1-tm* telomeres serves a different purpose than Rif2. It will be interesting to examine the significance of Sir3 at Rap1-depleted telomeres.

## Materials and methods

### Yeast strains and plasmids

Standard yeast media and growth conditions were used (*Sherman, 2002*; *Lundblad and Struhl, 2010*). Yeast strains used in this study are listed in *Supplementary file 1A*. The artificial VII-L telomere was created essentially as previously described (*Gottschling et al., 1990*) by transforming restriction-digested pVII-L URA-TEL plasmid into yeast. To obtain different versions of the VII-L telomere, this plasmid was modified by replacing the original 81 bp of telomeric repeats with oligonucleotides containing wild-type or *tlc1-tm* telomeric sequence of different length (*Supplementary file 1B*). Myc and protA C-terminal epitope-tagged proteins were created by integrative transformation of 13 copies of the human c-myc (Myc) epitope or the Staphylococcal protein A IgG binding domain.

### Southern blotting

Southern blots to detect native telomeres were performed essentially as previously described (*van Mourik et al., 2018*). Yeast genomic DNA was isolated using a Wizard Genomic DNA Purification Kit (Promega) and digested with XhoI. For each sample, 4 µg of digested genomic DNA, along with 1.25 ng of BsmAI-digested pYt103 (*Askree et al., 2004*), was separated on a 0.8% (w/v) agarose gel and transferred to a Hybond-N+ membrane (GE Healthcare). The membrane was hybridized to telomere-specific digoxigenin-labeled probe (wild-type probe: 5′-CACCACACCCACACACCACA CCCACA-3′; *tlc1-tm* mutant probe: 5′-ACCACACCACACCACACACACCACAC-3′). For detection of the artificial VII-L telomere, a similar procedure was performed, except that 6 µg yeast genomic DNA was digested with EcoRV and the membrane was hybridized at 42°C with a digoxigenin-labeled probe complementary to *URA3* sequence. Unless otherwise mentioned (i.e., *Figures 1C and 3B*), all strains were propagated for at least 100 generations before Southern blot analysis.

### Chromatin immunoprecipitation and quantitative PCR (ChIP-qPCR)

ChIP-qPCR was performed essentially as previously described (*Graf et al., 2017*). For protein A ChIP, IgG Sepharose 6 Fast Flow beads (GE Healthcare) were used. For Myc ChIP, nProtein A Sepharose 4 Fast Flow beads (GE Healthcare) were used; after preclearing, 9 µl anti-c-Myc Monoclonal antibody (Clontech/Takara) were added to each sample. qPCR was performed using a LightCycler 480 II (Roche) and SYBR-Green (Thermo Scientific) detection with an annealing temperature of 60°C (40 cycles).

Primers are listed in *Supplementary file 1C*. Measured Cq values were corrected to input and graphs were created using R. All strains were propagated for ~50 generations before ChIP-qPCR analysis.

## Electrophoretic mobility shift assay (EMSA)

The recombinant *S. cerevisiae* Rap1 full-length protein was expressed in *Escherichia coli* BL21 cells as previously described (*Wahlin and Cohn, 2000*). The ability of Rap1 to bind the *tlc1-tm* telomeric sequences was assessed by EMSA with two double-stranded (ds) oligonucleotides containing representative sequences; Mut-1 5'-*GTCATACGTCACAC*TGTGGTGTGTGGTGTGGTGTGTGGTGGTGGTGTGTGT-3' (37 bp) and Mut-2 5'-*GTCATACGTCACAC*TGTGGTGTGTGTGGTGTGTGGATTTGGTGTGTGG-3' (34 bp). The respective G-rich forward strands and C-rich reverse strands were annealed in 1 mM Tris-HCl, pH 8.0, 0.1 mM MgCl$_2$ by boiling for 2 min and slowly cooling down to room temperature. The correct annealing was guided by the 14 nt non-telomeric region indicated in italics. A ds-oligonucleotide with wild-type telomeric sequence was used as the positive binding control: 5'-TGTGGTGTGTGGGTGTGTG-3' (19 bp). The 5' ends of ds-probes were radioactively labeled with [γ-$^{32}$P]-ATP using T4 polynucleotide kinase (New England Biolabs), purified on Illustra Microspin G-25 columns (GE), and diluted in 10 mM Tris-HCl, pH 7.5. In binding assays, 10 fmol of labeled probe was mixed with 1.5 µg competitor mix (0.5 µg each of sheared *E. coli* DNA [~250 bp], salmon sperm DNA, and yeast t-RNA) in 1× binding buffer (10 mM Tris-HCl, pH 7.5, 7 mM MgCl$_2$, 8% glycerol), and varying amounts (~0.1–0.4 µg) of Rap1 protein extract, in a total of 15 µl. The binding reaction was incubated at 25°C for 15 min and then loaded onto 4% polyacrylamide gels (29:1 acrylamide:bis-acrylamide) and run in 1× TBE (89 mM Tris-borate, 2 mM EDTA, pH 8.0), 150 V at 4°C. Radioactive signals from dried gels were visualized with a Typhoon FLA 9500 biomolecular imager (GE Life Sciences). Quantification of signals in the respective lanes was performed using the ImageQuant TL software. Bands were automatically detected with the settings: minimum slope 15, median filter 10, % maximum peak 0. Background subtraction was performed using the 'rolling ball' method with a radius of 200. The fraction of bound probe (%) was calculated in each lane as (shifted signal)/(total signal of shifted and unshifted bands).

## iSTEX assay

iSTEX was performed essentially as previously described (*Strecker et al., 2017*), except that the starting P$_{GALL}$-*EST1* strain contained the *tlc1-tm* allele. This strain was then transformed with a PCR product containing the wild-type *TLC1* allele and immediately grown on media containing glucose, which shut off expression of *EST1*. Successfully transformed cells were cultured in YPD media and eventually arrested in late G1 phase by the addition of alpha factor (Sigma-Aldrich). Cells were washed and resuspended in YPGal to induce the expression of wild-type telomerase, allowing the extension of *tlc1-tm* mutant telomeres during a single cell division.

## Liquid culture senescence assay

Senescence assays in liquid culture were performed essentially as previously described (*van Mourik et al., 2016*).

## TERRA and URA3 level analysis by quantitative reverse transcription and PCR

TERRA levels were measured essentially as previously described (*Graf et al., 2017*), while URA3 and ACT1 levels were measured by one-step RT-PCR with specific primers (*Supplementary file 1C*) following the standard protocol of the RNeasy Mini kit (QIAGEN). All strains were propagated for at least 100 generations before analysis.

## Acknowledgements

We thank K Paeschke and L Wanders for critical reading of the manuscript; S Pandey and O Rosas Bringas for help with data analysis; and HG Kazemier for technical assistance. FRRB was supported by a CONACYT scholarship. SS was supported by an EMBO Short-Term Fellowship to visit the lab of B Luke to perform TERRA measurements and to gain experience with ChIP-qPCR experiments, and we thank B Luke and M Graf for their technical assistance. MC was supported by grants from the Carl

Trygger Foundation, the Erik Philip-Sörensen Foundation, and the Royal Physiographic Society in Lund. Work in the laboratory of MC was supported by a Vidi grant from the Netherlands Organization for Scientific Research.

## Additional information

### Funding

| Funder | Grant reference number | Author |
|---|---|---|
| Nederlandse Organisatie voor Wetenschappelijk Onderzoek | Vidi grant:864.12.002 | Michael Chang |
| Carl Tryggers Stiftelse för Vetenskaplig Forskning | | Marita Cohn |
| Erik Philip-Sörensen Foundation | | Marita Cohn |
| Royal Physiographic Society in Lund | | Marita Cohn |
| Consejo Nacional de Ciencia y Tecnología | PhD scholarship | Fernando Rodrigo Rosas Bringas |
| European Molecular Biology Organization | Short-Term Fellowship | Sonia Stinus |

The funders had no role in study design, data collection and interpretation, or the decision to submit the work for publication.

### Author contributions

Fernando Rodrigo Rosas Bringas, Formal analysis, Funding acquisition, Investigation, Supervision, Writing - original draft, Writing - review and editing; Sonia Stinus, Formal analysis, Funding acquisition, Investigation, Writing - original draft, Writing - review and editing; Pien de Zoeten, Investigation, Writing - review and editing; Marita Cohn, Funding acquisition, Investigation, Writing - review and editing; Michael Chang, Conceptualization, Formal analysis, Funding acquisition, Investigation, Project administration, Supervision, Writing - original draft, Writing - review and editing

### Author ORCIDs

Fernando Rodrigo Rosas Bringas (ib) http://orcid.org/0000-0002-7810-0613
Michael Chang (ib) http://orcid.org/0000-0002-1706-3337

### Decision letter and Author response
Decision letter https://doi.org/10.7554/eLife.74090.sa1
Author response https://doi.org/10.7554/eLife.74090.sa2

## Additional files

### Supplementary files
• Supplementary file 1. Supplementary tables. (A) Yeast strains used in this study. (B) Wild-type and mutant oligo sequences used to replace the VII-L telomere. (C) qPCR primers.

• Transparent reporting form

### Data availability
All data generated or analysed during this study are included in the manuscript and source data files.

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
