## [Editor Report]

The study clarifies the role of Rif2 in telomere homeostasis and how cells can extend telomeres and control senescence in the absence of the Rap1 binding to telomeres. The possibility of coping with telomere sequence modification through flexibility and redundancy of capping proteins is of general interest in terms of telomere evolution.

---

## [Decision Letter]

**Decision letter after peer review:**

Thank you for submitting your article "Rif2 protects Rap1-depleted telomeres from MRX-mediated degradation in *Saccharomyces cerevisiae*" for consideration by *eLife*. Your article has been reviewed by 3 peer reviewers, one of whom is a member of our Board of Reviewing Editors, and the evaluation has been overseen by Jessica Tyler as the Senior Editor. The following individual involved in review of your submission has agreed to reveal their identity: Stéphane Marcand (Reviewer #2).

Essential revisions:

– It seems clear that "recruitment" to MRX, as shown in rad50∆ mutants, is necessary for Rif2 protection in tlc1-tm mutants, but if Rif2 is able to bind DNA, is it possible to generate a DNA-binding-compromised mutant of Rif2, to see whether it can still fulfill its biological function? This will allow you to determine up to which point Rif2 binding to MRX is occurring at the DNA end to which Rif2 is being recruited independently. Alternatively, you could check whether Rif2 co-IPs with MRX (even though a negative result may still be not completely conclusive for technical reasons, it still may strengthen your arguments).

– To complement tlc1-tm rif2∆ cells with a Rif2 mutant specifically unable to interact with Rap1 (e.g. rif2-L44R-V45E-E347R from Shi et al., 2013 Cell 153:1340), but still able to inhibit MRX would validate the model that Rif2 recruitment is Rap1-independent.

–You did not investigate Sir3, even though in principle it is an additional player able to bind to Rap1-less telomeres. Could you explain at least what would be the expectation, or better yet, perform experiments with sir3 mutants to check whether its behavior is similar to that of Rif2?

– Figure 6B : cdc13-1 tlc1-tm cells do not exhibit the smear observed in rif2 TLC1-tm but instead show a typical profile of type II-like recombination that suggests that Cdc13 has indeed a role in capping TLC1-tm telomeres. Are cdc13-1 tlc1-tm cells viable in the absence of Rad52?

– Figure 7D: How much of the reduction in Rif2 ChIP signal in rad50∆ cells is due to telomere shortening? Adding tel1∆ and tel1∆ tlc1-tm cells as additional controls can address this issue.

Other comments:

– Emphasize what is how your conclusions differ from prior work, considering the large amount of information existing on this topic and the redundancy of Rap1 function with other proteins.

– Figure 5 is referred to on p. 8 out of order.

– Size markers are missing in Figure 1B.

– Figure 1E, Rap1 still binds the mutant telomeric sequences. An estimation of the remaining affinity and its comparison with scramble sequences would help to interpret this result.

– page 12, line 272-273: the comparison with type II survivors is a bit obscure, since type II survivors are MRX-dependent. This paragraph can be improved or simplified.

– The sentence "We find that Rap1, Rif2, and the Ku complex work in parallel to prevent telomere degradation, and absence of all three at telomeres causes lethality" should be amended. Did you refer to WT or tlc1-tm cells ? In WT cells, Rif2 is rather a regulator of the telomerase than a capping protein and the concerted role of Rap1, KU and Cdc13 is well documented. In tlc1-tm cells, the decreased availability of Rap1 reveals (or creates?) a capping function of Rif2 that acts in parallel with KU and Cdc13.

– Rif2 interacts with DNA in vitro. Is this relevant for Rif2 recruitment in the absence of Rap1? Can this hypothesis be explored? And is there a remaining telomere specificity in this recruitment?

– A question emerging is whether Rif2 would have a role on DSB repair in general. There are standard NHEJ and HR assays using HO to see whether rif2- cells show DSB repair defects. Wouldn't this be a prediction if Rif2 apparently interacts and work together with non-specific telomeric proteins such as the Yku complex or MRX? Alternatively they can check Rad52 foci accumulation as an indirect measurement. Authors should at least discuss this possibility and future directions in this context.

– The term "degradation" used to describe telomeres in rif2 tlc1-tm cells does not seem appropriate in the absence of further characterization. The retention of DNA in the well could be also related to the presence of replication intermediates or telomere entanglement. The smear is indeed partially abrogated by the deletion of Rad52. What are the effects of deletion of EXO1?

– The use of 6Y' in the ChIP experiments makes comparison with TELVII-Lmut difficult. Another X- telomere would have been more appropriate. Explain minimally why it was not used.

– Line 399-400: The reference to Bonetti et al., 2009 is not appropriate. On the contrary, this study was suggesting a role for Sae2 in the formation of the 3'-overhang, a role that was later questioned by Geli's lab (Hardy et al., Nat Comm 2014).

– The vocabulary used to describe the decreased binding of Rap1 to tlc1-tm telomeric repeats should be standardized taking into account that (1) Rap1 binding is not totally abolished (2) telomeres (except TELVII-L mut) retain RAP1 binding sites in their most proximal part as well as on interstitial telomeric repeats in Y' telomeres. It seems therefore more correct to speak of a decreased level of Rap1 rather than loss or lack of Rap1.

– The role of recombination in the homeostasis of tlc1-tm telomeres should be discussed because stochastic recombination events or processing of intermediates could be at the origin of variations of the telomere profiles between isogenic tlc1-tm clones. In this line, the figure 5-Figure Supp2 shows a clear difference in the telomere profiles of tlc1-tm and tlc1-tm rad52 cells but this is not commented in the text. Similarly, the large heterogeneity of X-telomeres is not mentioned though it could be due to de novo acquisition of Y' sequences at least in some cases. Finally, Rad52 is required for the growth of rif2 tlc1-tm cells.

---

## [Author Response]

Essential revisions:– It seems clear that "recruitment" to MRX, as shown in rad50∆ mutants, is necessary for Rif2 protection in tlc1-tm mutants, but if Rif2 is able to bind DNA, is it possible to generate a DNA-binding-compromised mutant of Rif2, to see whether it can still fulfill its biological function? This will allow you to determine up to which point Rif2 binding to MRX is occurring at the DNA end to which Rif2 is being recruited independently. Alternatively, you could check whether Rif2 co-IPs with MRX (even though a negative result may still be not completely conclusive for technical reasons, it still may strengthen your arguments).

It would indeed be very interesting to test a DNA-binding-compromised mutant of Rif2, but none has been described, nor can be easily predicted by sequence analysis. We feel that an extensive search for such a mutant would take months of work (at minimum), and is therefore beyond the scope of our current study. Regarding the co-IP experiment, we are unclear as to what it would accomplish because Rif2 is already known to physically interact with both the Rad50 and Xrs2 subunits of the MRX complex.

– To complement tlc1-tm rif2∆ cells with a Rif2 mutant specifically unable to interact with Rap1 (e.g. rif2-L44R-V45E-E347R from Shi et al., 2013 Cell 153:1340), but still able to inhibit MRX would validate the model that Rif2 recruitment is Rap1-independent.

This is an excellent suggestion. We cloned *RIF2* or *rif2-L44R,V45E,E347R* under the control of the *RIF2* promoter into the pRS315 plasmid. We transformed the plasmids, along with the vector control, into *rif2∆* and *rif2∆ tlc1-tm* cells. After passaging the cells for ~100 generations, we found that expression of wild-type *RIF2* partially recovered the telomere profile of *rif2∆ tlc1-tm* cells (full recovery likely requires additional generations), and fully suppressed the growth defect (new Figure 7—figure supplement 1). However, expression of *rif2-L44R,V45E,E347R* could do neither, indicating that the ability to interact with Rap1 was important for Rif2 protection of *tlc1-tm* telomeres, but it is also possible that the L44R, V45E, and E347R mutations disrupt the ability of Rif2 to protect *tlc1-tm* telomeres in a manner independent of the Rap1-Rif2 interaction. This is discussed at the end of the Results section.

–You did not investigate Sir3, even though in principle it is an additional player able to bind to Rap1-less telomeres. Could you explain at least what would be the expectation, or better yet, perform experiments with sir3 mutants to check whether its behavior is similar to that of Rif2?

We deleted *SIR3* in wild-type, *tlc1-tm*, and *tlc1-476A* strains (new Figure 4—figure supplement 1). We saw no detectable change in cell growth. Telomere length is longer in *sir3∆ tlc1-tm* than *tlc1-tm* cells, but there is no detectable difference in telomere profile of *tlc1-476A* and *sir3∆ tlc1-476A*. These observations have now been added to the end of the Discussion, which concludes with:

“Thus, the presence of Sir3 at *tlc1-tm* telomeres serves a different purpose than Rif2. It will be interesting to examine the significance of Sir3 at Rap1-depleted telomeres.”

– Figure 6B: cdc13-1 tlc1-tm cells do not exhibit the smear observed in rif2 TLC1-tm but instead show a typical profile of type II-like recombination that suggests that Cdc13 has indeed a role in capping TLC1-tm telomeres. Are cdc13-1 tlc1-tm cells viable in the absence of Rad52?

We deleted *RAD52* in *cdc13-1 tlc1-tm* strains; the resulting triple mutant is viable but grows very poorly. The type II-like telomere profile of *cdc13-1 tlc1-tm* cells is also dependent on *RAD52*. These observations are shown in the new Figure 6—figure supplement 1. We conclude that Rad52 is important for maintaining *cdc13-1 tlc1-tm* telomeres.

– Figure 7D: How much of the reduction in Rif2 ChIP signal in rad50∆ cells is due to telomere shortening? Adding tel1∆ and tel1∆ tlc1-tm cells as additional controls can address this issue.

The main conclusion from Figure 7D is that Rif2 ChIP signal is lost in *rad50∆ tlc1-tm* cells compared to *rad50∆* cells. Since both *rad50∆* and *rad50∆ tlc1-tm* cells have similarly short telomeres (Figure 6B), the signal loss is not due to a difference in telomere length. If we compare Rif2 ChIP signal for WT (Figure 4A) and *rad50∆* (Figure 7D), we do see a reduction in signal at the VII-L telomere for *rad50∆*, but not at Y’ telomeres, so this effect seems specific for the VII-L telomere. At present, we do not have an explanation for the relatively high Rif2 ChIP signal detected for the VII-L telomere, and an experiment with *tel1∆* and *tel1∆ tlc1-tm* is unlikely to solve this mystery. Such an experiment would be further complicated by the fact that Tel1 promotes MRX accumulation at DNA ends (PMID: 26901759), so a *tel1∆* mutation may also affect Rif2 ChIP signal independently of telomere length differences.

Other comments:– Emphasize what is how your conclusions differ from prior work, considering the large amount of information existing on this topic and the redundancy of Rap1 function with other proteins.

We believe that the changes we have made in the revised manuscript have addressed this issue.

– Figure 5 is referred to on p. 8 out of order.

This has been corrected.

– Size markers are missing in Figure 1B.

A different blot with size markers is now included.

– Figure 1E, Rap1 still binds the mutant telomeric sequences. An estimation of the remaining affinity and its comparison with scramble sequences would help to interpret this result.

We have now included a calculation of the percentage of bound DNA for the wild-type and mutant oligos. Scrambled sequences would not be relevant to test because a huge amount of competitor DNA is used in the EMSA, as stated in the Materials and methods:

“10 fmol of labelled probe was mixed with 1.5 µg competitor mix (0.5 µg each of sheared *E. coli* DNA (~250 bp), salmon sperm DNA, and yeast t-RNA)”.

Thus, the binding we detect is highly specific, as expected since Rap1 is known to be very sequence-specific in its binding. If Rap1 had even a low non-sequence-dependent affinity to DNA, we would not be able to see a mobility shift of the labelled probe because it would be outcompeted by the non-labelled competitor DNA.

– page 12, line 272-273: the comparison with type II survivors is a bit obscure, since type II survivors are MRX-dependent. This paragraph can be improved or simplified.

We have now noted that the *rad50∆ tlc1-tm* and *rad50∆ rif2∆ tlc1-tm* strains in Figure 6B are unlikely type II survivors because Rad50 is required for type II survivor formation.

– The sentence "We find that Rap1, Rif2, and the Ku complex work in parallel to prevent telomere degradation, and absence of all three at telomeres causes lethality" should be amended. Did you refer to WT or tlc1-tm cells ? In WT cells, Rif2 is rather a regulator of the telomerase than a capping protein and the concerted role of Rap1, KU and Cdc13 is well documented. In tlc1-tm cells, the decreased availability of Rap1 reveals (or creates?) a capping function of Rif2 that acts in parallel with KU and Cdc13.

We have amended the sentence as suggested. The new sentence reads: “Rif2 and the Ku complex work in parallel to prevent *tlc1-tm* telomere degradation; *tlc1-tm* cells lacking Rif2 and the Ku complex are inviable. We have also deleted the sentence in the first paragraph of the Discussion that read:

“Thus, Rap1, Rif2, and the Yku complex perform separate tasks at telomeres that are together essential for telomere capping and cell viability.”

– Rif2 interacts with DNA in vitro. Is this relevant for Rif2 recruitment in the absence of Rap1? Can this hypothesis be explored? And is there a remaining telomere specificity in this recruitment?

It would indeed be interesting to test whether Rif2’s ability to interact directly with DNA is important for its recruitment to Rap1-depleted telomeres. Unfortunately, a DNA-binding-defective mutant of Rif2 has not been described. Figure 8D of Cassani et al., 2016, examines Rif2 binding to presumably nontelomeric dsDNA while Figure S3 of Hailemariam et al., 2019, examines Rif2 binding to telomeric DNA. Although the exact experimental conditions differ, the results suggest that binding of Rif2 to telomeric and non-telomeric DNA is similar.

– A question emerging is whether Rif2 would have a role on DSB repair in general. There are standard NHEJ and HR assays using HO to see whether rif2- cells show DSB repair defects. Wouldn't this be a prediction if Rif2 apparently interacts and work together with non-specific telomeric proteins such as the Yku complex or MRX? Alternatively they can check Rad52 foci accumulation as an indirect measurement. Authors should at least discuss this possibility and future directions in this context.

Rif2 indeed has a role in DSB repair, as has been documented by the Longhese lab. We described what is known in the Results section (first paragraph of subsection entitled “Rif2 protects *tlc1-tm* telomeres by inhibiting the MRX complex”). Specifically, we note that “Rif2 enhancement of Rad50 ATPase activity limits MRX-mediated tethering of DNA ends during double-strand break repair (Cassani et al., 2016).” In the Discussion, we further note that “the role of Rif2 in DSB repair is more readily detected in the absence of Sae2 (Marsella et al., 2021).”

– The term "degradation" used to describe telomeres in rif2 tlc1-tm cells does not seem appropriate in the absence of further characterization. The retention of DNA in the well could be also related to the presence of replication intermediates or telomere entanglement. The smear is indeed partially abrogated by the deletion of Rad52. What are the effects of deletion of EXO1?

We use the term “telomere degradation” because it was previously used by Elizabeth Blackburn’s group to describe such a telomere profile (e.g. PMIDs 10733598 and 14742705). Deletion of *EXO1* cannot suppress the *rif2∆ tlc1-tm* phenotype (see Author response image 1; strains were obtained from the dissection of an *exo1∆/EXO1 rif2∆/RIF2 tlc1-tm/TLC1* diploid).

**Author response image 1. sa2fig1:** 

– The use of 6Y' in the ChIP experiments makes comparison with TELVII-Lmut difficult. Another X- telomere would have been more appropriate. Explain minimally why it was not used.

We chose to examine 6 Y’ telomeres to complement the artificial VII-L telomere because (i) it allowed us to examine several telomeres at once, and (ii) it has been reported by the Petes lab that telomeres lacking subtelomeric elements (like the artificial VII-L telomere) are more similar to Y’ telomeres than X telomeres, at least in terms of length regulation (PMID 10430581).

– Line 399-400: The reference to Bonetti et al., 2009 is not appropriate. On the contrary, this study was suggesting a role for Sae2 in the formation of the 3'-overhang, a role that was later questioned by Geli's lab (Hardy et al., Nat Comm 2014).

We thank the reviewer for pointing this out to us. We have now changed the reference (PMID 34311383). This reference is a recent review article (also by Bonetti and colleagues) that discusses the interplay between Sae2 and Rif2 at DSBs and telomeres.

– The vocabulary used to describe the decreased binding of Rap1 to tlc1-tm telomeric repeats should be standardized taking into account that (1) Rap1 binding is not totally abolished (2) telomeres (except TELVII-L mut) retain RAP1 binding sites in their most proximal part as well as on interstitial telomeric repeats in Y' telomeres. It seems therefore more correct to speak of a decreased level of Rap1 rather than loss or lack of Rap1.

In the revised manuscript, we have ensured that we always say that Rap1 levels are decreased or depleted at *tlc1-tm* telomeres (not lost).

– The role of recombination in the homeostasis of tlc1-tm telomeres should be discussed because stochastic recombination events or processing of intermediates could be at the origin of variations of the telomere profiles between isogenic tlc1-tm clones. In this line, the figure 5-Figure Supp2 shows a clear difference in the telomere profiles of tlc1-tm and tlc1-tm rad52 cells but this is not commented in the text. Similarly, the large heterogeneity of X-telomeres is not mentioned though it could be due to de novo acquisition of Y' sequences at least in some cases. Finally, Rad52 is required for the growth of rif2 tlc1-tm cells.

We agree that Rad52-mediated recombination is involved in the maintenance of *tlc1-tm* telomeres. We had already mentioned the effect of Rad52 on the telomere profile and growth of *rif2∆ tlc1-tm* cells (Figure 5C and 5D). We now also note that “*rad52∆ tlc1-tm* cells also have longer telomeres than *tlc1-tm* cells (Figure 5C), indicating that HR is important for maintaining *tlc1-tm* telomeres.” The issue of telomere heterogeneity is already discussed in the Results section. In short, the heterogeneity observed in *rad50∆ tlc1-tm* and *rad50∆ rif2∆ tlc1-tm* cells is still present in the absence of Rad52, indicating that recombination is not responsible. We speculate that the heterogeneity “may be related to a previous observation that the very tip of the telomere has unique sequence features that affect telomere capping and length regulation (Grossi et al., 2001).” In contrast, the heterogeneity seen in *cdc13-1 tlc1-tm* is dependent on Rad52 (new Figure 6—figure supplement 1). We believe that the revised manuscript now more adequately discusses the role of recombination in the maintenance of *tlc1-tm* telomeres.